# A Rhizobacterium, *Streptomyces albulus* Z1-04-02, Displays Antifungal Activity against Sclerotium Rot in Mungbean

**DOI:** 10.3390/plants11192607

**Published:** 2022-10-04

**Authors:** On-Uma Ruangwong, Kaewalin Kunasakdakul, Sompong Chankaew, Kitsada Pitija, Anurag Sunpapao

**Affiliations:** 1Department of Entomology and Plant Pathology, Faculty of Agriculture, Chiang Mai University, Mueang, Chiang Mai 50200, Thailand; 2Innovative Agriculture Research Center, Faculty of Agriculture, Chiang Mai University, Chiang Mai 50200, Thailand; 3Department of Agronomy, Faculty of Agriculture, Khon Kaen University, Khon Kaen 40002, Thailand; 4Perkin Elmer Co., Ltd., 290 Soi 17, Rama 9 Rd., Bangkapi, Huay Kwang, Bangkok 10310, Thailand; 5Agricultural Innovation and Management Division (Pest Management), Faculty of Natural Resources, Prince of Songkla University, Hatyai, Songkhla 90110, Thailand

**Keywords:** actinobacteria, antibiosis, cell wall degrading enzymes, *Sclerotium*, volatile organic compounds

## Abstract

Sclerotium rot causes damping-off and stem rot in seedlings and mature mungbeans, which negatively impacts cultivation. The use of a rhizobacterium to control soil-borne diseases is an alternative method to the excess use of synthetic fungicides; therefore, this study aims to screen rhizosphere actinobacteria with fungicidal activities against *Sclerotium rolfsii*, the pathogen that causes sclerotium rot in mungbeans. Primary screening showed that the *Streptomyces* sp. isolate Z1-04-02 displayed the highest effectiveness against *S. rolfsii* in dual culture plates, with a percentage inhibition of 74.28%. An assay containing enzymes that degrade cell walls, of the cell-free culture filtrate (CF) of Z1-04-02, showed that the activities of chitinase and *β*-1,3-glucanase were 0.0209 and 1.0210 U/mL, respectively, which was significantly higher than that of the control (media alone). The cell-free CF of Z1-04-02, incubated at 37 °C and 100 °C, using agar well diffusion, effectively inhibited the growth of *S. rolfsii* with inhibition percentages of 37.78% and 27.78%, respectively. Solid-phase microextraction (SPME) was applied to trap volatiles released from Z1-04-02 and gas chromatography–mass spectrometry (GC/MS); volatile antifungal compounds were tentatively identified as bicyclic monoterpene (1R)-(-)-myrtenal. The application of the cell-free CF, and the spore suspension of Z1-04-02, showed disease severity indexes (DSIs) of 12.5% and 8.25%, respectively, which were significantly lower than those showing inoculation by *S. rolfsii* alone. The identification of this strain by morphology, biochemistry tests, and 16s rDNA sequences revealed that Z1-04-02 was *Streptomyces albulus*. This finding revealed that *S. albulus* Z1-04-02 displayed diverse fungicidal activities against *S. rolfsii*, and it has the potential to act as a biological control agent in terms of inhibiting sclerotium rot in mungbeans.

## 1. Introduction

Rhizosphere soil is an important source of beneficial microorganisms, especially rhizosphere microorganisms, which have been studied for their beneficial effects on plant growth and their roles as biocontrol agents [1,2,3]. Actinobacteria are fungal-like microorganisms of the order Actinomycetales. The structure of actinobacteria consists filaments in a network, and actinobacteria can produce spores or conidia. Actinobacteria are Gram-positive and they mostly inhabit soil, utilizing soil substrates and the decaying organic matter in soil. *Streptomyces* is a genus of the family Streptomycetaceae [4], which has been widely studied for its contributions as a biological control agent (BCA) that works against plant diseases [5,6]. Different strains of *Streptomyces* can produce cell wall degrading enzymes (CWDEs), such as chitinase and β-1,3-glucanase, volatile organic compounds (VOCs), and antimicrobial metabolites, which are responsible for the suppression of plant pathogens [7,8,9,10,11,12]. Furthermore, some *Streptomyces* strains can induce disease resistance [13,14]. 

Several strains of *Streptomyces* have been widely applied in agriculture to control plant fungal diseases, and they have diverse antimicrobial abilities. For instance, *Streptomyces alfalfa* produced various extracellular hydrolytic enzymes and diffusible antifungal metabolites that work against *Fusarium oxysporum* f.sp. *vasinfectum*, the pathogen of fusarium wilt in cotton [15]. *Streptomyces griseorubiginosus* regulated plant defense enzyme activities and induced sugarcane smut resistance by regulating stress-related enzyme activities in sugarcane [16]. *Streptomyces sichuanensis* produced siderophore and antifungal metabolites that work against *F. oxysporum* f.sp. *cubense* tropical race 4, the causal agent of fusarium wilt in banana [17]. *Streptomyces* sp. produced cellulolytic, chitinolytic, and proteolytic extracellular enzymes that inhibit *Macrophomena phaseolina* and *Rhizoctonia solani* in *Phaseolus vulgaris* [18]. Furthermore, *Streptomyces* sp. AN090126 produced and secreted antimicrobial metabolites into culture filtrates, and they produced volatile antifungal compounds that work against plant pathogens [19].

The mungbean (*Vigna radiata*) is a plant species of the legume family, and it is mainly grown in the tropical and subtropical areas of East Asia and Southeast Asia, including China, India, Pakistan, Thailand, Indonesia, and the Philippines [20]. As with most economic crops, global mungbean production often encounters serious problems that can cause a reduction in yield. The major problem is that it is susceptible to southern blight or sclerotium rot caused by *Sclerotium rolfsii* [21]. In Thailand, the cultivation of mungbean is distributed throughout the country, and most cultivation areas are located in northeastern Thailand (according to the Office of Agricultural Economics, Thailand). 

In Thailand, mungbeans also suffer from sclerotium rot, even though the use of synthetic fungicides effectively controls sclerotium rot [22,23]; however, the use of high doses of fungicides might cause the development of chemical resistance [24], and it may also cause harmful side effects that impact human health [25]. *Streptomyces* strains have exhibited strong antibiosis against plant diseases, and therefore, the use of actinobacteria from the rhizosphere to control soil-borne pathogens is an alternative method to control this disease. This study aimed to find the effective strains of the rhizobacterium *Streptomyces*, which have diverse fungicidal abilities, to suppress *S. rolfsii* and to reduce disease severity in mungbean crops.

## 2. Results

### 2.1. Antifungal Ability of Streptomyces against Sclerotium rolfsii

All isolates of actinobacteria effectively inhibited the mycelial growth of *S. rolfsii* by dual culture assay, with the inhibition percentage ranging from 36.67% to 74.29%. The *Streptomyces* sp. strain Z1-04-02 showed the highest inhibition percentage against *S. rolfsii*, with an inhibition percentage of 74.29%, which is significantly higher than that of other strains (Figure 1); therefore, the *Streptomyces* strain Z1-04-02 was selected for further study.

### 2.2. Cell Wall Degrading Enzyme Activities of Z1-04-02 Cell-Free CF

The ability to suppress *S. rolfsii* may result from CWDE activities; therefore, enzyme assays of chitinase and *β*-1,3-glucanase were conducted through the cell-free CF of Z1-04-02. The CWDEs activities of chitinase and *β*-1,3-glucanase in the cell-free CF of Z1-04-02 were 0.0209 and 1.0210 U/mL, respectively, which are significantly higher than that observed in glucose yeast malt broth (GYMB) alone (Figure 2).

### 2.3. Cell-Free CF of Z1-04-02 Inhibited Growth of Sclerotium rolfsii

To confirm that the cell-free culture filtrate (CF) of Z1-04-02 contained antifungal metabolites and/or heat stable antifungal metabolites, agar well diffusion was conducted using the cell-free CF incubated at an ambient temperature and 100 °C. The results show that the cell-free CF, incubated at both the ambient temperature and 100 °C, suppressed the fungal growth of *S. rolfsii* in PDA plates (Figure 3). The inhibition percentages of the Z1-04-02 cell-free CF, incubated at an ambient temperature and 100°C, were 37.78% and 27.78%, respectively (Figure 3).

### 2.4. GC/MS Profiling of Volatiles Emitted by Z1-04-02

The ability to suppress fungal growth may be the result of antifungal metabolites and/or VOCs emitted by the Z1-04-02 strain, and therefore, solid-phase microextraction (SPME) was conducted to trap volatiles and tentatively identify them through gas chromatography–mass spectrometry (GC/MS). GC/MS profiling revealed four terpene compounds, namely 2,6-dimethyl-1,3,6-heptatriene (C_9_H_14_), (1R)-(-)-myrtenal (C_10_H_14_O), 3,3-dimethyl-6-methylenecyclohexene (C_9_H_14_), and 7-propylidene-bicyclo[4.1.0]heptane (C_10_H_16_), in volatiles of Z1-04-02. Details of each compound are presented in Table 1, and the major peaks of the mass spectra are presented in Figure 4.

### 2.5. Sclerotium Rot Disease Suppression

The results show that the application of the Z1-04-02 spore suspension and cell-free CF prevented damping-off, caused by *S. rolfsii*, in mungbean seedlings (Figure 5). The application of the cell-free CF and spore suspension of Z1-04-02 showed disease severity indexes (DSIs) of 12.5% and 8.25%, respectively, which are significantly lower than those showing inoculation by *S. rolfsii* alone (75%). Seven days after inoculation, damping-off occurred in the control mungbean seedlings (inoculation by *S. rolfsii* alone), whereas it did not occur in those receiving treatment (Figure 5).

### 2.6. Identification of Z1-04-02

A biochemistry test on Z1-04-02 revealed that it hydrolyzes starch and utilizes urea. Furthermore, Z1-04-02 oxidizes glucose, lactose, sucrose, and mannitol. Z1-04-02 is Gram positive. The morphology of Z1-04-02 showed a dark gray aerial spore mass and light-yellowish substrate mycelia. This strain grows well in GYMA when incubated in a temperature range of 30–35 °C.

The 16S rDNA of Z1-04-02 is about 1200 bases long. A BLASTN search revealed that this strain is 99.69% identical to that of *Streptomyces albulus*. The maximum likelihood tree of the 16S rDNA of Z1-04-02, and related species, as shown in the National Center for Biotechnology Information’s (NCBI) database, showed that the Z1-04-02 strain of the current study is similar to *S. albulus* (Figure 6). The DNA sequence of *S. albulus* Z1-04-02 was deposited in GenBank and it acquired the accession number LC670616. Based on the morphology, biochemistry test, and molecular study of 16S rDNA, the *Streptomyces* sp. strain Z1-04-02 was identified as *S. albulus*.

## 3. Discussion

In the current study, the most effective strain of actinobacteria, *S. albulus*, was isolated and identified based on morphology, a biochemistry test, and the nucleotide sequences of 16S rDNA. *Streptomyces albulus* displayed a strong antibiosis mechanism both in vitro and in vivo against *S. rolfsii*, the pathogen causing mungbean stem rot. According to our consistent results from both assessments, it was revealed that *S. albulus* has the potential to be a candidate for BCA in restricting stem rot in mungbeans due to its capacity for the production of antifungal compounds, CWDEs, and VOCs.

One of the most effective abilities of *Streptomyces* is the production of extracellular metabolites that were produced and released into a medium to restrict fungal growth in PDA plates [10,26]. The antifungal metabolites released by *Streptomyces* species may consist of extracellular CWDEs, chitinase, and β-1,3-glucanase [27], as well as heat stabilizing antifungal metabolites [10,28] and VOCs [29,30]. As observed in this study, using a dual culture assay, *S. albulus* Z1-04-02 effectively inhibited the growth of *S. rolfsii* in dual culture plates. This result suggests that *S. albulus* Z1-04-02 produced antifungal metabolites in GYMA media, which caused the restriction of fungal growth.

Cell wall degrading enzymes composed of chitinase and β-1,3-glucanase are responsible for degrading the cell wall components, chitin and β-glucan, into small molecules respectively; they are also involved in antifungal activities [31,32,33]. Hydrolytic enzymes produced by *Streptomyces* caused abnormalities in fungal morphology, as observed by scanning electron microscope [10]. Some strains of *Streptomyces* can produce and release CWDEs into media, as observed with the enzyme assay [27,28]. For instance, *S. cavourensis* SY224 produces chitinase and β-1,3-glucanase, thus inhibiting the growth of *Colletotrichum gloeosporioides*, and causing anthracnose in pepper [28]. In this study, we found that *S. albulus* Z1-04-02 also produces chitinase and β-1,3-glucanase, as observed with the enzyme assay. This ability may be due to an antagonistic activity in *S. albulus* Z1-04-02 that works against fungal pathogens.

The current study shows that the inhibitory effect of *S. albulus* Z1-04-02 against the mycelial growth of *S. rolfsii* may be due to its capacity to produce and release heat stabilizing antifungal compounds. A similar result was observed in *S*. *cavourensis*, which showed a heat stabilizing metabolite that worked against *C. gloeosporioides* [28]. Furthermore, Wonglom et al. [10] found that *S. angustmyceticus* NR8-2 produced a heat stabilizing antifungal compound that is responsible for suppressing the growth of *Colletotrichum* sp. and *Curvularia lunata*. In the present study, heat-treated cell-free CF showed a lower inhibition percentage than that of untreated cell-free CF, and this may be due to the heat treatment deactivating some antifungal compounds in the cell-free CF; however, we did not observe metabolite components in this study other than the observed CWDEs and VOCs for the biological control of stem rot in mung beans.

Among the four major VOCs found in this study, a bicyclic monoterpene myrtenal has been observed in several plants with antimicrobial activity [34]. Myrtenal and its derivatives have been widely studied via bioassay. A novel myrtenal-based compound has been shown to have strong antifungal abilities against plant disease pathogens, including *F. oxysporum* f. sp. *cucumerinum*, *Physalospora piricola*, *Alternaria solani*, *Cercospora arachidicola*, and *Gibberella zeae* [35]. Some myrtenal derivatives exhibit favorable antifungal activity, for instance, myrtenal-derived 2-(p-methylbenzoyl)-1,2,4-triazole-3-thione 71a strongly inhibited the growth of *Bipolaris maydis* and *P. piricola* [36]. Interestingly, *S. albulus* Z1-04-02 is the first species in this genus to produce this volatile compound. A different strain of *S. albulus*, NJZJSA2, has been reported to produce dominant VOCs, including 4-methoxystyrene and 2-pentylfuran, which display antifungal activities against *Sclerotinia sclerotiorum* and *F. oxysporum* [37]. In the current study, we found that *S. albulus* Z1-04-02 produced four dominant VOCs that differed from other strains; therefore, different strains of *Streptomyces* species may produce different VOCs with diverse mechanisms.

Several *Streptomyces* species have been widely studied as BCAs against plant diseases. *Streptomyces* sp. was found to be a good potential biocontrol agent for use against wood decay fungus, *Gloeophyllum trabeum* [38]. *Streptomyces palmae* PC12 showed strong antifungal activities against rice blast disease fungi and increased plant growth [39]. Recently, wuyiencin, a nucleoside antibiotic producing *S. albulus* CK-15, has been reported to be effective against cucumber powdery mildew [40]. The results from our study are in agreement with previous studies, as *S. albulus* Z1-04-02 exhibits diverse fungicidal activity against the growth of *S. rolfsii* and reduces the disease severity of stem rot in mungbeans. This study aimed to find an effective *Streptomyces* strain to inhibit a soil-borne pathogen, *S. rolfsii*, and to reduce the disease severity caused by this pathogen; therefore, an in vivo test showed that the application of *S. albulus* Z1-04-02 reduced stem rot in mungbeans compared with a control, thus suggesting that it is an effective potential biological control agent against stem rot caused by *S. rolfsii*.

## 4. Materials and Methods

### 4.1. Sources of Actinobacteria and Pathogen

Six strains of rhizosphere actinobacteria, namely, *Kitasatospora nipponensis* KM6-4 [41], *S. angustmyceticus* NR8-2 [10], and *Streptomyces* spp. TH23-7 [42], T3-04-02, Z1-04-02, and Z2-04-01, and *S. rolfsii* were obtained from the Culture Collection of the Pest Management Department, Faculty of Natural Resources, Prince of Songkla University, Thailand. All strains of actinobacteria were cultured in glucose yeast malt agar (GYMA), and *S. rolfsii* was cultured in potato dextrose agar (PDA) at 28 ± 2 °C for 3 days before use in this study.

### 4.2. Dual Culture Assay

To test the antifungal ability of actinobacteria against *S. rolfsii*, a dual culture assay was conducted on the PDA plates. Each strain of actinobacterium was one-line streaked on PDA plates and they were incubated at 28 ± 2 °C. After 5 days of incubation, a plug (0.5 cm diameter) from the edge of a 2-day-old colony of *S. rolfsii* was placed on each tested plate on the opposite side. The plates without actinobacteria served as controls. The experiment was composed of three replicates and was repeated twice. The colony radii of *S. rolfsii* were measured after 5 days of incubation in the dual tests. The colony radii of *S. rolfsii* were measured and the percentage inhibition was calculated with the following formula:(1)$$\mathrm{Percentage}\;\mathrm{of}\;\mathrm{inhibition}=\frac{R1-R2}{R1}\times 100,$$
where *R*1 is the colony radius of *S. rolfsii* in the absence of antagonist bacteria, and *R*2 is the colony radius of *S. rolfsii* in the presence of antagonist bacteria [43].

### 4.3. Bacterial Cultivation and Enzyme Assay

We hypothesized that actinobacteria can produce and secrete extracellular metabolites that are responsible for the restriction of fungal growth. The most effective strain of actinobacterium was subjected to cultivation in GYMB, and it was incubated at an ambient temperature (28 ± 2 °C) for 7 days. The cultured broth was filtrated using 0.2 μm filter paper and used as a cell-free CF. Cell wall degrading enzymes, including chitinase and β-1,3-glucanase, were assayed using the 3,5-dinitrosalicylic acid (DNS) method [44]. Colloidal chitin and laminarin (Sigma-Aldrich, St. Louis, MO, USA) were used as a substrate for the chitinase β-1,3-glucanase assay. The products of the enzyme assay (reducing sugar) released in the reaction mixtures were measured with a UV5300 UV/VIS spectrophotometer (METASH, Shanghai, China) at 550 nm and 575 nm for *β*-1,3-glucanase and chitinase, respectively. Each enzyme assay was performed in three replicates and was repeated twice.

### 4.4. Agar Well Diffusion

We tested whether the antifungal metabolites in the cell-free CF of the selected actinobacterium contained heat stabilizing compounds with antifungal abilities. The cell-free CF of the selected actinobacterium was incubated at 37 °C and 100 °C for 1 h and it was subjected to the agar well diffusion method. This method reveals the fungal growth inhibition on PDA plates by the cell-free CF of the actinobacterium. The treatment was composed of sterile distilled water (DW), GYMB, and cell-free CF. Each treatment was composed of three replicates and the experiment was repeated twice. The tested plates were incubated at an ambient temperature for 2 days, and the colony radii of *S. rolfsii* were measured and converted to inhibition percentages using the following formula:Percentage inhibition = [(C − T)/C] × 100,(2)
where C indicates the radial growth of *S. rolfsii* from the center of the control well and T indicates the radial growth of *S. rolfsii* from the center of the cell-free CF well [9].

### 4.5. Solid-Phase Microextraction–Gas Chromatography–Mass Spectrometry (SPME-GC/MS)

To detect the VOCs found in the effective actinobacterium, SPME–GC/MS was conducted [45]. The selected actinobacterium was cultured in GYMA in a 20 mL chromatography vial (PerkinElmer, Waltham, MA, USA) and incubated at ambient temperature for 7 days. SPME (DVB/CAR/PDMS) fiber was exposed to the vapor phase above the actinomycete for 30 min in a culture tube [46,47,48]. Then, the adsorbent fiber was inserted into the injection port of a Clarus model 690 gas chromatograph (PerkinElmer, Waltham, MA, USA) coupled to the model SQ8 mass-selective detector, equipped with Elite-5MS (5% phenylmethylpolysiloxane with a 30 m × 250 μm ID × 0.25 μm film thickness). The column temperature was set at an initial temperature of 60 °C and increased at a rate of 7 °C/min to a final temperature of 200 °C. Purified helium gas was used as the carrier gas at a flow rate of 1 mL/min. Electron impact (EI) mass spectra were collected at 70 eV ionization voltage over the range of m/z 45–550. The VOCs produced by the selected actinomycete were tentatively identified through a computer search of the National Institute of Standard and Technology (NIST, v17, 2014) Mass Spectral Library Search Chromatogram.

### 4.6. In Planta Test

To test the effect of actinobacteria in reducing stem rot disease caused by *S. rolfsii*, a pot experiment was conducted. Twenty mungbean plants were grown in sterile soil in a 35 × 52 × 4 cm pot with capacity of each well 3 × 3 × 4 cm. Inoculation of *S. rolfsii* was conducted by directly placing one agar plug of *S. rolfsii* onto the basal stem of each mungbean seedling (seedling stage, 5–10 cm height with 2 leaves), whereas the cell-free CF or spore suspension of the actinobacteria was applied via spraying onto the basal stem of the mungbean in a volume of 10 mL for each plant. The cell-free CF of selected actinobacteria was diluted at a 1:1 ratio (DW: cell-free CF), whereas the spore suspension of the actinobacteria was prepared by harvesting spores from 10-day-old colonies and adjusting the concentration to 10^8^ spore/mL with sterile distilled water. The experiment was performed using a randomized complete block design (RCBD) with four treatments: (1) treated with 10 mL DW (control); (2) mungbean inoculation with *S. rolfsii* alone; (3) mungbean inoculation with *S. rolfsii* and 10 mL cell-free CF; (4) mungbean inoculation with *S. rolfsii* and 10 mL spore suspension of the actinobacteria. Five mungbean plants were inoculated in accordance with each method and the experiment was conducted three times. The tested mungbean plants were incubated at ambient temperature and the disease progress with stem rot and wilting was observed after 7 days of inoculation. The disease scores were determined by the method previously described by Chiang et al. [49], with some modifications, based on assessing the external symptoms of the plant (0 = no symptoms, 1 = lesion development without wilting, 2 = lesion development with up to two wilted leaves, 3 = more than two wilted leaves, 4 = plant damping-off and dead). The disease scores were converted to a disease severity index (DSI) as follows:(3)$$\mathrm{DSI}\;\left(\%\right)=\frac{\sum (\mathrm{Scale}\times \mathrm{Amount}\;\mathrm{of}\;\mathrm{plants})}{\mathrm{Maximum}\;\mathrm{level}\times \mathrm{Total}\;\mathrm{number}\;\mathrm{of}\;\mathrm{plants}}\times 100$$

### 4.7. Biochemistry Test and Identification of Actinobacterium

Biochemical tests of features such as starch hydrolysis, utilization of urea, and oxidation and fermentation of glucose, lactose, sucrose, and mannitol were conducted using the method of Reddy et al. [50]. Colony growth and the macroscopic and microscopic features of the selected actinobacteria were observed by stereomicroscope (Leica S8AP0, Leica Microsystems, Wetzlar, Germany) and a compound microscope (Leica DM750, Leica Microsystems, Wetzlar, Germany). The morphological characteristics of the spore chains of the selected *Streptomyces* species were observed using a scanning electron microscope [41]. The most effective strain was cultured in GYMA for 24 h and subjected to direct PCR as a DNA template. Amplification of the 16s rDNA was conducted using a BIORAD T100^TM^ Thermal Cycler (Bio-Rad, Hercules, CA, USA). A portion of the 16s rDNA of the selected actinobacteria was amplified by PCR using a 27F forward (5′ AGAGTTTGATCMTGGCTCAG 3′) and 1389R reverse (5′ ACGGGCGGTGTGTACAAG 3′) primer pair. The PCR reaction was prepared sequentially in a 50 μL reaction tube containing 10 pmol of each primer, 2× DreamTaq Green PCR Master Mix (Thermo Fisher Scientific, Waltham, MA, USA), and one colony of the actinobacteria as a DNA template. An initial denaturation was set to 3 min at 94 °C followed by 35 cycles of denaturation for 1 min at 94 °C, annealing for 1 min at 60 °C, and extension for 3 min at 72 °C, with a final extension step of 10 min at 72 °C. The PCR product was then observed using a 1% agarose gel electrophoresis. The 16s rDNA gene region was sequenced by the WARD MEDIC sequencing service (Ward Medic, Bangkok, Thailand). The sequences were BLASTN searched (NCBI) and aligned using program MEGA X [51]. A phylogenetic tree of a 16s rDNA gene sequence of the selected actinobacteria was constructed using the maximum likelihood method with 1000 bootstrap replications.

### 4.8. Statistical Analysis

Significant differences between pathogen growth, CWDE activities, and disease severity were subjected to a one-way analysis of variance (ANOVA). Tukey’s test and Student’s *t* test were used to analyze statistically significant differences.

## 5. Conclusions

A rhizobacterium, *S. albulus* Z1-04-02, displayed strong fungicidal activities against *S. rolfsii*, the pathogen of stem rot in mungbeans. The core antibiosis mechanisms are the production of CWDEs, antifungal metabolites, and VOCs, which successfully suppressed the growth of *S. rolfsii*. The application of *S. albulus* reduced stem rot in mungbean seedlings; therefore, *S. albulus* has the potential to be a good biocontrol agent for the suppression of sclerotium rot in mungbean seedling. Effect of *S. albulus* for suppressing sclerotium rot in all stage of growth should be further verified. Moreover, development of formulation of *S. albulus* to use in the filed need to be further clarified in future.

## Figures and Tables

**Figure 1 plants-11-02607-f001:**
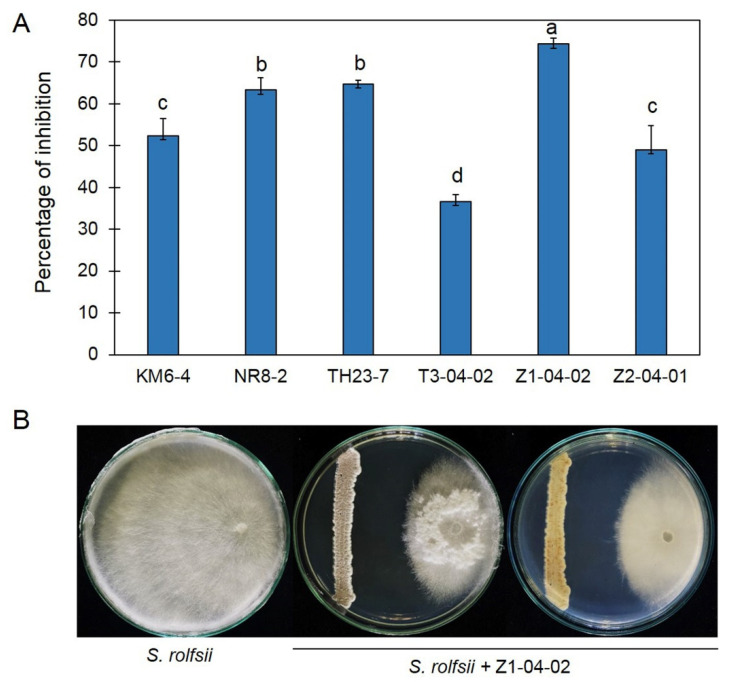
Percentage inhibition of *Streptomyces* spp. against *Sclerotium rolfsii* by dual culture assay (**A**); Z1-04-02 effectively inhibited the growth of *S. rolfsii* in the tested plate (**B**). Data are means ± SD; error bars indicate standard deviation (SD). Different letters indicate significant differences according to Tukey’s test (*p* < 0.05).

**Figure 2 plants-11-02607-f002:**
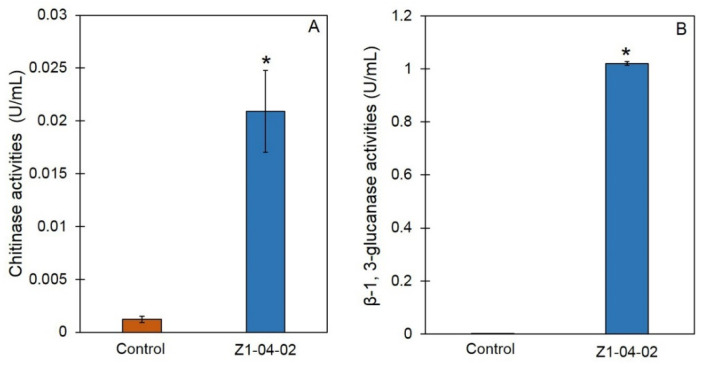
Cell wall degrading enzyme activities of chitinase (**A**) and β-1,3-glucanase (**B**). Data are means ± SD; error bars indicate standard deviation (SD). Asterisks indicate significant differences according to Student’s *t* test (*p* < 0.05).

**Figure 3 plants-11-02607-f003:**
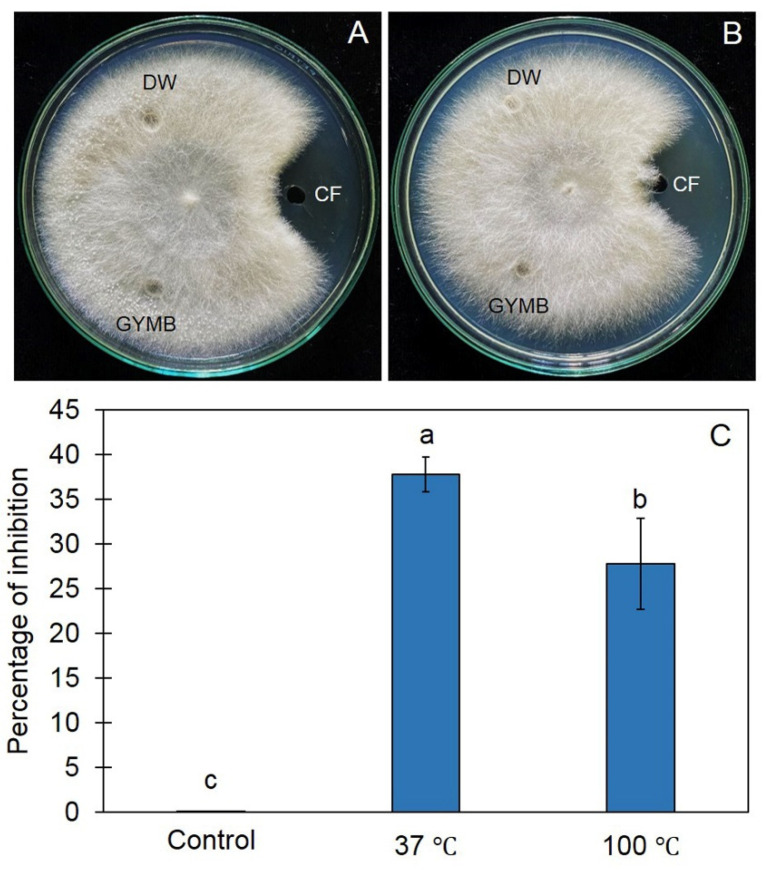
Effect of temperature on the antifungal ability of the cell-free culture filtrate (CF) incubated at 37 °C (**A**), 100 °C (**B**), and the percentage inhibition of the cell-free CF (**C**). Different letters indicate significant differences according to Tukey’s test (*p* < 0.05).

**Figure 4 plants-11-02607-f004:**
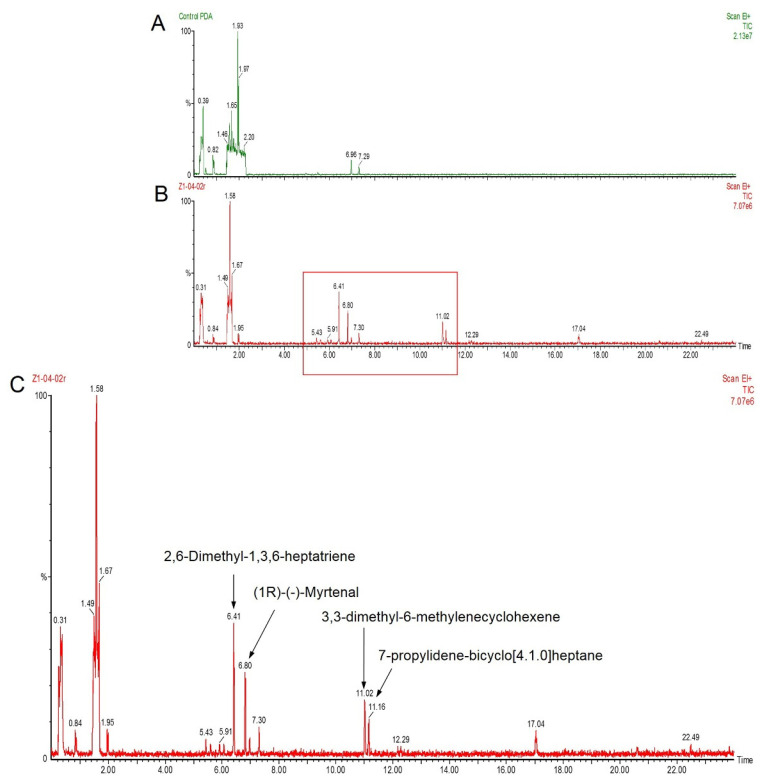
GC/MS profiling of volatiles emitted by Z1-04-02: mass spectrum of PDA alone (**A**), mass spectrum of Z1-04-02 (**B**), and qualitative volatile compounds of Z1-04-02 (**C**).

**Figure 5 plants-11-02607-f005:**
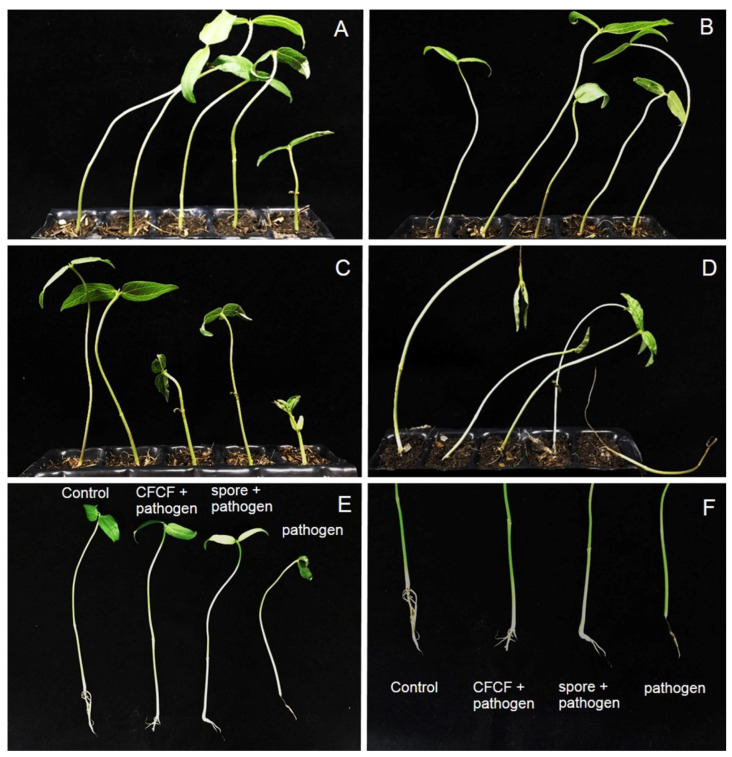
In planta test of the cell-free culture filtrate (CFCF), and the spore suspension of *Streptomyces* Z1-04-02, to combat sclerotium rot in mungbean seedlings: treatment with distilled water (**A**), CFCF (**B**), spore suspension (**C**), *Sclerotium rolfsii* only (**D**), symptom development in the mungbean plant (**E**), and a zoomed-in view of symptoms (**F**).

**Figure 6 plants-11-02607-f006:**
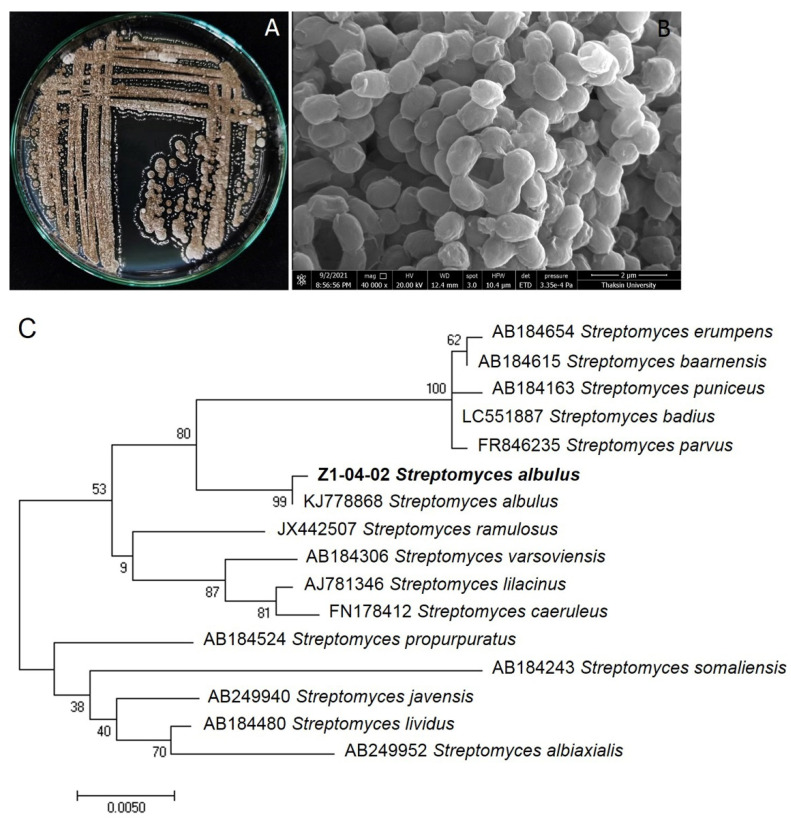
Morphology of *Streptomyces albulus* Z1-04-02 in GYMA (**A**), morphology of spore shape observed by SEM (**B**), and maximum likelihood tree of the 16s rDNA of *S. albulus* Z1-04-02 strain and related species with 1000 bootstrap replications (**C**). Bold letters indicate the sample used in this study.

**Table 1 plants-11-02607-t001:** Volatile compounds produced by *Streptomyces* sp. Z1-04-02, identified through GC/MS analysis.

RT (min)	Compounds	%Match	Formula	MW	Peak Area
6.41	2,6-Dimethyl-1,3,6-heptatriene	89	C_9_H_14_	122	76,249
6.80	(1R)-(-)-Myrtenal	87	C_10_H_14_O	150	44,937
11.02	3,3-dimethyl-6-methylenecyclohexene	86	C_9_H_14_	122	36,671
11.17	7-propylidene-bicyclo[4.1.0]heptane	89	C_10_H_16_	136	23,318

## Data Availability

Not applicable.

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
