# Peer review of "A Rhizobacterium, Streptomyces albulus Z1-04-02, Displays Antifungal Activity against Sclerotium Rot in Mungbean"

_plants, 2022, doi:10.3390/plants11192607_

Round 1

Reviewer 1 Report

Looking for the natural substances  that act  as fungicides to reduction of plant pathogen development is very needed. Therefore studies presented in paper concerned possibility the use of Rhizobacterium and  Streptomyces albulus Z1-04-02  for control of  Sclerotium Rot in Mungbean are interested and justified. However, that the results are confirm and were useful for  agricultural practice it should be precisely evaluate the conditions of obtained preparations use (dose, term plant developmental phase, number of treatments, efficiency of a preparation …). On this basis I have following suggestions and comments to the reviewed manuscript:

11. It should be more precisely characterized of mungbean plants during the applying of spraying (developmental phase, height of plants, number of leaves).

22. On which of basis  the date/term of spraying was selected?

33. Description of methods presented in division 4.6 is not understand. I am not sure which dose was applied 10 or 20 ml per plant:

·         lines 293-297: Inoculation of S. rolfsii was conducted by directly placing one agar plug of S. rolfsii onto the basal stem of each mungbean seedling, whereas the cell-free CF or spore suspension of the actinobacteria was applied via spraying onto the basal stem of the mungbean in a volume of 20 mL for each plant.

·         lines 300-304: The experiment was performed using a randomized complete block design (RCBD) with four treatments: 1) treated with 10 mL DW (control); 2) mungbean inoculation with S. rolfsii alone; 3) mungbean inoculation with S. rolfsii and 10 mL cell-free CF; 4) mungbean inoculation with S. rolfsii and 10 mL spore  suspension of the actinobacteria.

44. Why such big dose was applied only one time and whether this dose was not too big for these small plants?

55. There is lack of pot’s parameters like volume, dimensions/size.

66. Why the evaluation of pathogens infection was performed only 1 time from inoculation. Whether in the further developmental phases plants are not infected by S. rolfsii ?

77. Why plants in pots were not cultivated to the full maturity?

88. Obtaining of positive results in the laboratory often does not proofed in the field experiments. In regard to early harvest of plants in these studies in the summary should be planned pot experiments conducted during the whole ontogenesis from sowing to the seed harvest instead of field experiments. And only on the base of these results should be decided whether the field experiments are needed.

Author Response

Reviewer 1

Looking for the natural substances that act as fungicides to reduction of plant pathogen development is very needed. Therefore studies presented in paper concerned possibility the use of Rhizobacterium and Streptomyces albulus Z1-04-02 for control of Sclerotium Rot in Mungbean are interested and justified. However, that the results are confirm and were useful for  agricultural practice it should be precisely evaluate the conditions of obtained preparations use (dose, term plant developmental phase, number of treatments, efficiency of a preparation …). On this basis I have following suggestions and comments to the reviewed manuscript:

  1. It should be more precisely characterized of mungbean plants during the applying of spraying (developmental phase, height of plants, number of leaves).

Answer: We have revised in section 4.6.

  1. On which of basis the date/term of spraying was selected?

Answer: We applied by spraying for the first date followed by inoculation.

  1. Description of methods presented in division 4.6 is not understand. I am not sure which dose was applied 10 or 20 ml per plant:
  • lines 293-297: Inoculation of S. rolfsii was conducted by directly placing one agar plug of S. rolfsii onto the basal stem of each mungbean seedling, whereas the cell-free CF or spore suspension of the actinobacteria was applied via spraying onto the basal stem of the mungbean in a volume of 20 mL for each plant.
  • lines 300-304: The experiment was performed using a randomized complete block design (RCBD) with four treatments: 1) treated with 10 mL DW (control); 2) mungbean inoculation with S. rolfsii alone; 3) mungbean inoculation with S. rolfsii and 10 mL cell-free CF; 4) mungbean inoculation with S. rolfsii and 10 mL spore suspension of the actinobacteria.

Answer: We have revised as 10 mL

  1. Why such big dose was applied only one time and whether this dose was not too big for these small plants?

Answer: Only 10 mL was not a big dose to treat with one seedlings and the concentration was diluted with a ratio of 1:1. Moreover, we applied only one time per inoculation to test in this concentration of Streptomyces could suppress the disease or not.

  1. There is lack of pot’s parameters like volume, dimensions/size.

Answer: We have added in materials and methods section 4.6.

  1. Why the evaluation of pathogens infection was performed only 1 time from inoculation. Whether in the further developmental phases plants are not infected by S. rolfsii?

Answer: We applied only one time per inoculation to test in this concentration of Streptomyces could suppress the disease or not. Period of incubation for 7 days, control plants showed severe case by inoculation, and we have collected the results. For further study in all stage of mungbean, we have added in summary.

  1. Why plants in pots were not cultivated to the full maturity?

Answer: Because, we would like to develop as formulation to test from seedling to mature stage in the field, therefore for primary screening of effective microorganisms was conducted in seedlings stage. For further study in all stage of mungbean, we have added in summary.

  1. Obtaining of positive results in the laboratory often does not proofed in the field experiments. In regard to early harvest of plants in these studies in the summary should be planned pot experiments conducted during the whole ontogenesis from sowing to the seed harvest instead of field experiments. And only on the base of these results should be decided whether the field experiments are needed.

Answer: We have added this information into summary.

Reviewer 2 Report

A Rhizobacterium, Streptomyces albulus Z1-04-02, Displays Antifungal Activity Against Sclerotium Rot in Mung bean

General Comments:

The manuscript entitled, A Rhizobacterium, Streptomyces albulus Z1-04-02, Displays Antifungal Activity Against Sclerotium Rot in Mung bean is a well-structured manuscript with relevant justifications. The manuscript appears to be in order sound and possesses good presentation clarity. The article is worth publishing however, A few basic comments regarding the manuscript can be listed as under:

Major comments

·         Author can justify if MIC can be estimated along with inhibition zone?

·         Why did author not choose to take any positive control? May be fungicidal compound commonly used in plant disease and or Streptomyces albulus that have been earlier reported inhibitory activity against Sclerotium rolfsii

Minor comments

·         There are many typos, font, space, text justification, hyphenations etc. issues that needed to be addressed.

·         Kindly use acronym across the manuscript after it has been mentioned once: e.g., volatile organic compounds, cell wall degrading enzymes,

·         Kindly italicized the scientific names of the organisms (missing at many places)

·         Fusarium oxysporum can be written as F. oxysporum after its first mention. Follow same for all the scientific names

·         Please keep the value and the units together without spaces throughout the manuscript

·         Ln 143: Please spell out GYMA in first occurrence

·         Ln 167, 225: italicize In vivo and In vitro

Author Response

Reviewer 2

General Comments:

The manuscript entitled, A Rhizobacterium, Streptomyces albulus Z1-04-02, Displays Antifungal Activity Against Sclerotium Rot in Mung bean is a well-structured manuscript with relevant justifications. The manuscript appears to be in order sound and possesses good presentation clarity. The article is worth publishing however, A few basic comments regarding the manuscript can be listed as under:

Major comments

  • Author can justify if MIC can be estimated along with inhibition zone?

Answer: In agricultural field, evaluation effective microorganism prefer inhibition zone or percentage inhibition more than evaluate MIC.   

  • Why did author not choose to take any positive control? May be fungicidal compound commonly used in plant disease and or Streptomyces albulus that have been earlier reported inhibitory activity against Sclerotium rolfsii.

Answer: There are 2 reasons, first in a current agricultural system in Thailand avoid to use synthetic fungicide in the field, second according to national strategy of Thailand support the sustainable agriculture. Therefore, our research was selected to test with and without antagonistic microorganism to control plant disease which may suitable for current agricultural system in Thailand. 

Minor comments

  • There are many typos, font, space, text justification, hyphenations etc. issues that needed to be addressed.

Answer: We have carefully checked and revised throughout this manuscript.

  • Kindly use acronym across the manuscript after it has been mentioned once: e.g., volatile organic compounds, cell wall degrading enzymes,

Answer: We have revised throughout this manuscript.

  • Kindly italicized the scientific names of the organisms (missing at many places)

Answer: We have revised according to reviewer comments. We italicized scientific name of organisms but we did not italicized the name of disease such as sclerotium rot, fusarium rot. etc.

  • Fusarium oxysporum can be written as F. oxysporum after its first mention. Follow same for all the scientific names

Answer: We have revised throughout manuscript.

  • Please keep the value and the units together without spaces throughout the manuscript

Answer: We have revised according to reviewer comments in an appropriate places.

  • Ln 143: Please spell out GYMA in first occurrence

Answer: We have given full name of GYMA in Line 143. 

  • Ln 167, 225: italicize In vivo and In vitro

Answer: We have italicize in vivo and in vitro in line 167 and 225.

Round 2

Reviewer 1 Report

I would like to inform, that Authors sufficiently took due account of my  comments contained in the review.

Therefore this paper in the present form can be published in the Plants journal.